# Cryo-EM structure of ABCG5/G8 in complex with modulating antibodies

Hanzhi Zhang[1,4], Ching-Shin Huang [1,4], Xinchao Yu [2,4], Jonas Lee[1], Amit Vaish[2], Qing Chen[2], Mingyue Zhou[3], Zhulun Wang[1] & Xiaoshan Min [1✉]

The heterodimer of ATP-binding cassette transporter ABCG5 and ABCG8 mediates the excretion of sterols from liver and intestine, playing a critical role in cholesterol homeostasis. Here, we present the cryo-EM structure of ABCG5/G8 in complex with the Fab fragments from two monoclonal antibodies at 3.3Å resolution. The high-resolution structure reveals a unique dimer interface between the nucleotide-binding domains (NBD) of opposing transporters, consisting of an ordered network of salt bridges between the conserved NPXDFXXD motif and serving as a pivot point that may be important for the transport cycle. While mAb 11F4 increases the ATPase activity potentially by stabilization of the NBD dimer formation, mAb 2E10 inhibits ATP hydrolysis, likely by restricting the relative movement between the RecA and helical domain of ABCG8 NBD. Our study not only provides insights into the structural elements important for the transport cycle but also reveals novel epitopes for potential therapeutic interventions.

---

[1] Department of Therapeutics Discovery, Amgen Research, Amgen Inc., South San Francisco, CA, USA. [2] Department of Therapeutics Discovery, Amgen Research, Amgen Inc., Thousand Oaks, CA, USA. [3] Department of Cardiometabolic Disorders, Amgen Research, Amgen Inc., South San Francisco, CA, USA. [4] These authors contributed equally: Hanzhi Zhang, Ching-Shin Huang, Xinchao Yu. ✉email: xiaoshanm@gmail.com

The ATP-binding cassette (ABC) transporters utilize the energy from ATP hydrolysis to transport a variety of substrates across membranes and are found in all kingdoms of life[1]. In humans, there are 48 ABC transporters divided into 7 sub-families (A–G)[2,3]. While all ABC transporters consist of a pair of nucleotide-binding domains (NBDs) and a pair of transmembrane domains (TMDs), the subfamily G (ABCG) members possess a unique architecture in which the NBD is N-terminal to the TMD[4]. Among the ABCG family members, ABCG5/G8 is essential for pumping cholesterol and phytosterols outward across apical membranes of enterocytes and hepatocytes[5]. It serves as a gatekeeper of sterol transport and acts opposite to the Niemann–Pick C1-Like protein 1 (NPC1L1). NPC1L1 facilitates sterol influx across apical membranes of enterocytes from intestinal lumen and hepatocytes from bile canaliculus in the liver[6,7], whereas ABCG5/G8 mediates efflux of the sterols in the cells to allow proper absorption of cholesterol while restricting the absorption of structurally similar phytosterols[5]. People with loss-of-function variants of *ABCG5* or *ABCG8* develop sitosterolemia, an autosomal disease characterized by impaired ability to eliminate dietary sterols[5,8]. Sitosterolemic patients have considerably higher plasma levels of phytosterols, which can develop tendon xanthomas, and pose a high risk of cardiovascular disease. On the other hand, gain-of-function mutation variants of *ABCG5/G8* are associated with gallstone disease[9,10].

The core molecular structure of the ABCG5/G8 transporter shares similarities to other members of the ABC transporters. It is comprised of heterodimeric TMDs and heterodimeric NBDs, which is responsible for ATP hydrolysis. The crystal structure of the human ABCG5/G8 in a nucleotide-free state was solved recently at a modest resolution of 3.9 Å[11]. While the resolution limits analysis of the structure in terms of atomic detail, the structure revealed the overall architecture of the TMDs and NBDs of the unique ABCG family member and shed light on the coupling between TMDs and NBDs. Structures of another ABCG family member, ABCG2, in different transport states and with substrate or inhibitors bound, have been solved using cryo-electron microscopic (cryo-EM) techniques[12–15]. The atomic-resolution structures of ABCG2 provide a molecular understanding of the transport cycle of ABCG2 and its poly-specificity. With the availability of the crystal structure of ABCG5/G8, a comprehensive panel of human variants of each half transporter that alter transporter activity with different behaviors have been mapped to the structure[16]. However, key questions such as substrate specificity and ATP-driven cholesterol export remain unanswered. It remains unclear how the molecular mechanism operates whereby energy from ATP hydrolysis in the NBD is coupled to transport sterol substrates across cell membranes through the TMD. Furthermore, the mechanisms through which ABCG5/G8 transporter differentiates phytosterol molecules from cholesterol for efflux have not been identified. Phytosterols are generally present in human diets at quantities comparable with cholesterol; however, only approximately 5% of dietary phytosterols are absorbed in healthy individuals in contrast to approximately 50% absorption of dietary cholesterol[17,18].

Antibodies provide powerful tools to study the function and structure of membrane proteins[19]. For cryo-EM studies, addition of antigen-binding fragment (Fab) of antibody increases the size of the sample protein by ~47 kDa, which improves signal-to-noise-ratio and facilitates image alignment and three-dimensional (3D) reconstruction[20]. Moreover, antibodies that modulate protein functions are useful to explore the structural mechanism of proteins. Several antibodies or antibody fragments have been used to understand the function[21,22] or aid the structure determination of ABC transporters[22,23]. In this report, we generated highly specific antibodies against ABCG5/G8 and obtained a high-resolution cryo-EM structure of human ABCG5/G8 with the Fabs of these antibodies to a resolution of 3.3 Å. Structural analysis and biochemical characterization of the antibodies facilitate our understanding of the coupling mechanism between NBD and TMD and provide insight into the mechanism of allosteric regulation of the transporter activity.

## Results

**Protein preparation and identification of anti-ABCG5/G8 antibodies for structure determination.** Full-length ABCG5/G8 was purified using detergent *n*-dodecyl-β-D-maltoside (DDM), and its activity was confirmed by an ATPase assay (Fig. 1A). The ABCG5/G8 heterodimer has a relatively small molecular weight of ~150 kDa and possesses a twofold pseudosymmetry, which poses challenges for high-resolution structure determination using cryo-EM. To increase the molecular weight of the protein particle and facilitate image alignment, we generated a panel of antibodies to form complexes with human ABCG5/G8. Among the panel of antibodies, we screened for high-affinity and conformation-specific binders using ELISA assay. Two antibodies, monoclonal antibodies (mAbs) 2E10 and 11F4, were subjected for further biochemical and biophysical characterization. Both antibodies bind to ABCG5/G8 with affinities of around 100 pM, measured by surface plasmon resonance (SPR) (Fig. 1B, C). Epitope binning experiments by SPR revealed that mAbs 2E10 and 11F4 bind to distinctive epitopes on ABCG5/G8 (Fig. 1D). For cryo-EM sample preparation, we purified the ternary complex of ABCG5/G8 with the Fab of both mAbs 2E10 and 11F4 using size-exclusion chromatography and re-constituted the complex into saposin A-based nanodiscs before cryo-grid preparation and data collection (Fig. 1E and Supplementary Fig. 1).

**Cryo-EM structure of Fab-ABCG5/G8 complex.** The structure was solved with an overall resolution calculated to 3.3 Å out of 492,931 selected particles according to the gold-standard Fourier shell correlation (FSC) 0.143 criteria (Fig. 2A, B, Supplementary Fig. 2, and Table 1). The TMD and NBD of both half transporters and the variable domains of both Fab fragments are of excellent density (Supplementary Fig. 3). The constant domains of the Fab fragments are of lower resolution, but the quality of density is sufficient for complete model building (Supplementary Fig. 2D and Fig. 2). Moreover, we were able to perform de novo model building on the NBD of both ABCG5 and ABCG8. After rebuilding the NBD, we corrected several registry errors that were present in the previous reported crystal structure[11]. These include a loop at the N-terminus of ABCG8 from residues 28 to 56 that were mis-assigned as residues 23–44 in the crystal structure, and a segment from residues 306 to 362 in ABCG8, which were labeled as residues 306–355 in the crystal structure with mis-assigned registry throughout this segment (Fig. 2C). Guided by cryo-EM structure, we re-built these segments for the crystal structure model and the re-refined structure fits better to the electron density (Supplementary Fig. 4). Additionally, the excellent quality of the density in the cryo-EM structure allowed us to model in several disordered loops and improve side chain fitting throughout the structure.

Overall, the cryo-EM structure of ABCG5/G8 overlays well with the crystal structure with a root mean square deviation (RMSD) of 1.49 Å out of 1097 residues (Supplementary Fig. 5A). The TMD of the cryo-EM structure superimpose very well with the crystal structure TMD, with an RMSD of 0.98 Å out of 476 residues. Although the crystal structure was solved in detergent lipid bicelles and the cryo-EM structure was determined in lipid nanodiscs, there is high similarity between the two structures.

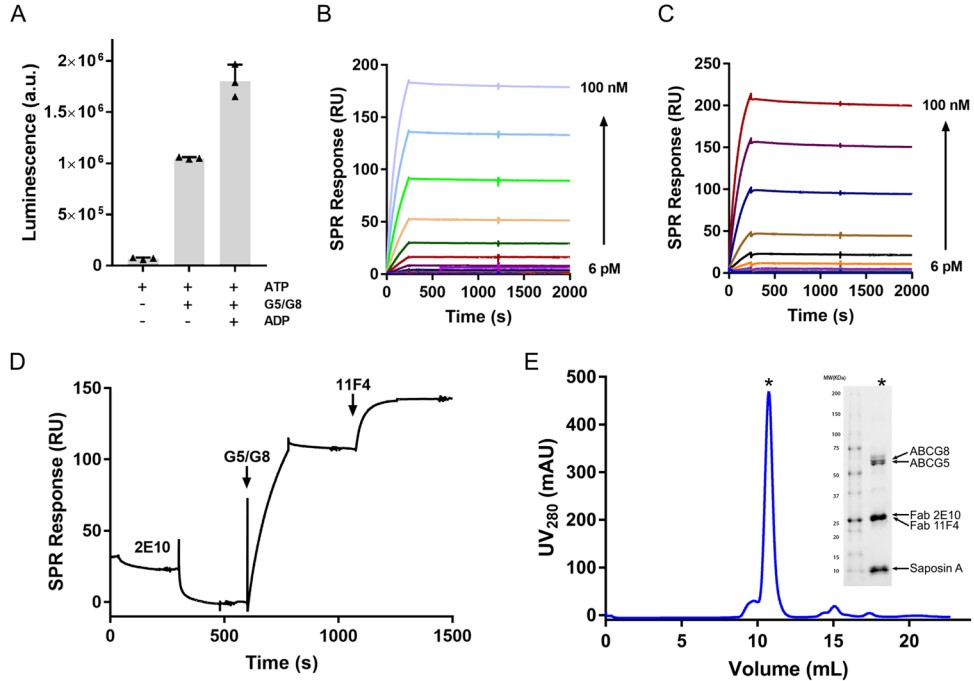

**Fig. 1 Characterization of ABCG5/G8 protein and its antibodies. A** ATPase activity of purified ABCG5/G8. **B** SPR measurement of the binding between mAb 2E10 and ABCG5/G8. **C** SPR measurement of the binding between mAb 11F4 and ABCG5/G8. In total, 15 mAb concentrations (6 pM–100 nM) were injected over immobilized ABCG5/G8 and yielded a $K_D$ value of 110 and 120 pM for mAb 2E10 (**B**) and 11F4 (**C**), respectively. **D** Epitope binning by SPR shows that mAb 2E10 and 11F4 have different epitopes for binding to ABCG5/G8. **E** Size-exclusion chromatogram of saposin A reconstituted ABCG5/G8 in complex with Fab fragments of mAbs 2E10 and 11F4. The peak fraction (designated by *) is characterized on SDS-PAGE.

These highly similar structures obtained from the orthogonal methods indicate that the observed conformation is likely to represent the transporter conformation in its native lipid environment. On the other hand, the NBD of the cryo-EM structure overlays with the crystal structure NBD with an RMSD of 1.62 Å out of 614 residues (Supplementary Fig. 5B, C), likely due to the poor electron density in the crystal structure at a lower resolution of 3.9 Å, which resulted in a few registry errors.

Like the crystal structure, the cryo-EM structure of ABCG5/G8 adopts an inward-facing conformation with two Fabs bound on the NBD of ABCG8. There was no density observed for either nucleotide, ATP or ADP, in the NBD or substrate in the TMD. While the NBDs of ABCG5 and ABCG8 packed against each other at the bottom, the nucleotide-binding sites (NBSs) are in an open conformation. The distance between the Lys92 of Walker A from ABCG5 and the invariant Ser214 in the signature motif from ABCG8 is about 23 Å, while the distance between the Arg111 of the Walker A motif from ABCG8 and the Ser194 from the signature motif of ABCG5 is about 19 Å (Fig. 2C). It was demonstrated that the NBS2, which consists of the ABCG5 Walker A motif, and opposing signature motif from ABCG8, is capable of mediating ATP hydrolysis, but the NBS1 formed by the ABCG8 Walker A motif and ABCG5 signature motif is degenerative[11]. Interestingly, in our cryo-EM structure, NBS2 shows well-resolved density while NBS1 is more disordered with less-defined side chains.

**A fixed NBD dimerization interface.** The registry-corrected segment from residues 306 to 362 in ABCG8 in our cryo-EM structure covers a conserved NPXDF motif that is important for ABCG family function[24]. Both crystal structure of ABCG5/G8 and cryo-EM structures of ABCG2 showed that the NPXDF motifs are located around the dimerization region between the two NBDs of each half transporter. Our cryo-EM structure with

improved resolution reveals a symmetric dimerization interface that is mediated by key residues from the NPXDF motifs. These key residues include two aspartate residues at the C-terminus of the short helical motif NPXDFXXD ($_{296}$NPFDFYMD$_{303}$ in ABCG5 and $_{316}$NPADFYVD$_{323}$ in ABCG8) forming two pairs of salt bridges with a positively charged arginine residue from the opposing half transporter, in a fashion similar to a zipper (Fig. 2C). The first salt bridge is formed between the side chain of Arg253 from ABCG5 and the side chains of two aspartic acid residues, Asp319 and Asp323, from ABCG8. The second pair is formed between the side chain of Arg273 from ABCG8 and the side chains of two aspartic acid residues, Asp299 and Asp303, in ABCG5 (Fig. 2C).

Sequence alignment of the NPXDFXXD motif suggests that the residues forming the salt bridges are conserved in the ABCG family and across species (Supplementary Fig. 6), comprising not only the two acidic residues in the NPXDFXXD motifs but also the arginine residue salt-bridging partner. One slight variation is between ABCG1/ABCG4 and ABCG2/ABCG5/ABCG8, where the positively charged residue changes to a lysine and is located two residues C-terminal to the conserved spot. It is possible that this lysine occupies similar space and fulfills the salt bridge interactions with the corresponding acidic residues in the opposing half transporter. Similar salt bridges are found in the ABCG2 structures solved in apo form[15], in inhibitor-bound form[13], and in ATP-bound conformation[14], which indicates that the salt bridges stay intact during the transport cycle. Thus, the salt bridges likely serve as a functional element that is equivalent to the C-terminal domain of ABC importers[25], which pinches the NBDs together and acts as a pivot point for the relative rotations between NBDs.

**The epitopes of the antibody 2E10 and 11F4.** Both Fab 2E10 and Fab 11F4 predominantly bind to the NBD of ABCG8

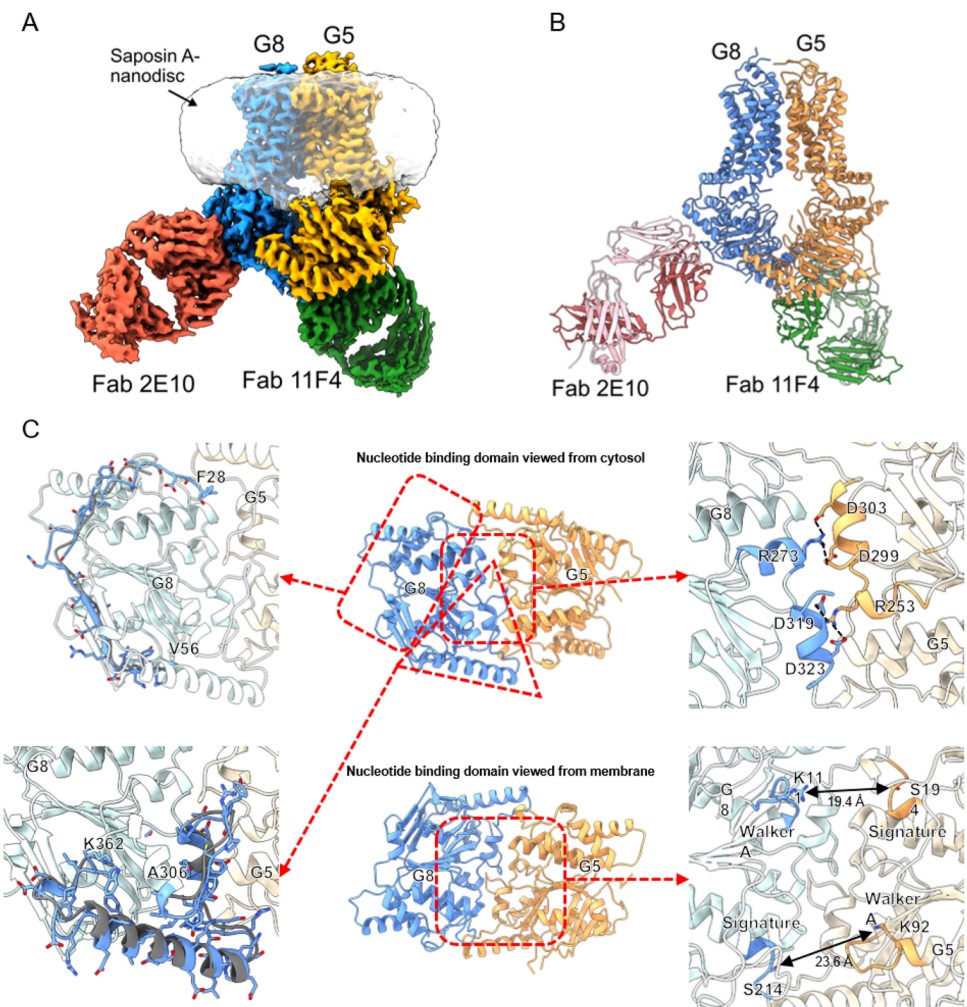

**Fig. 2 Structure of ABCG5/G8 in complex with Fab 2E10 and Fab 11F4. A** Cryo-EM map of ABCG5/G8 complexed with Fab 2E10 and Fab 11F4 in saposin A nanodisc. **B** Overall structure of ABCG5/G8 complexed with Fab 2E10 and Fab11F4. Heavy chain and light chain of Fab 2E10 are colored in red and light red, respectively. Heavy chain and light chain of Fab11F4 are colored in green and light green, respectively. **C** Structural details of ABCG5/G8 nucleotide-binding domains. Middle: the nucleotide-binding domains of ABCG5/G8 viewed from the cytosol (top) and membrane (bottom) sides. Top left: registry correction of amino acid residues, 28–56, in ABCG8 (blue, this study). Amino acid residues 24–44 from the crystal structure of ABCG5/G8 (PDB 5DO7) are shown in gray for comparison. Bottom left: registry correction of amino acid residues, 306–362, in ABCG8 (blue, this study). Amino acid residues 306–355 from the crystal structure of ABCG5/G8 (PDB 5DO7) are shown in gray for comparison. Top right: bottom view of ABCG5/G8 shows that the NPXDFXXD motifs from both ABCG5 and ABCG8 occupy the center location of a three-helix bundle and make the symmetric dimer interface between the NBDs. Bottom right: separation between ABCG5 signature motif and ABCG8 Walker A motif and between ABCG8 signature motif and ABCG5 Walker A motif.

(Fig. 2A, B). The NBD in ABC transporters is responsible for ATP hydrolysis, which provides the energy for the transfer of substrates across the membrane. We investigated whether these antibodies may affect the ATPase activity of ABCG5/G8. Surprisingly, mAb 2E10 inhibits the ATPase activity with an $IC_{50}$ of 49.4 nM, but mAb 11F4 potentiates ATPase activity with an $EC_{50}$ of 67.2 nM (Fig. 3A). To understand the structural basis of the effect of each antibody, we further analyzed the antibody epitope on ABCG5/G8.

Fab 2E10 interacts with both the RecA and the helical domains of the NBD from ABCG8 (Fig. 3B). The total buried surface area between Fab 2E10 and ABCG8 is ~1640 Å², with the heavy chain accounting for about three quarters of the interactions. On the Fab 2E10 side, all three complementarity-determining regions (CDRs), CDR1–3, from the heavy chain and CDR1 and CDR3 from the light chain are involved in the antigen recognition. Most noticeably, Tyr31, Ser55, His57, and Asn59 from the heavy chain form four hydrogen bonds with residues from the RecA domain

(Supplementary Fig. 7A, B). Asp101 from the heavy chain and Trp92 from the light chain form a salt bridge and a hydrogen bond, respectively, with the helical domain (Supplementary Fig. 7A, B). Cryo-EM studies of ABCG2 suggested that the helical domain, upon binding of ATP, rotates 35 degrees around the RecA domains to form the closed NBD dimer[14]. The fact that mAb 2E10 interacts with the RecA and helical domain simultaneously likely restricts this relative motion, thus hindering the completion of the ATP hydrolysis cycle. As a result, mAb 2E10 significantly reduces the ATPase activity of ABCG5/G8.

The epitope of Fab 11F4 is mostly on the NBD of ABCG8 and only includes a small footprint on the NBD of ABCG5 (Fig. 3C). The total buried surface area between Fab 11F4 and ABCG5/G8 is ~2000 Å², with the heavy chain and light chain contributing evenly to the interaction interface. All six CDRs from both the heavy chain and light chain are involved in interactions with ABCG5/G8. Fab 11F4 approaches ABCG5/G8 from the bottom of the heterodimer, away from the TMD, and close to the NBD

**Table 1 Cryo-EM data collection, refinement and validation statistics.**

| | ABCG5/G8 complexed with Fab 2E10 and 11F4 (EMDB-22443) (PDB 7JR7) |
|---|---|
| *Data collection and processing* | |
| Microscope | Titan Krios |
| Magnification | 130,000 |
| Voltage (kV) | 300 |
| Defocus range (μm) | −1.0 to −2.3 |
| Total electron dose (e⁻/Å²) | 47.5 |
| Exposure time (s) | 6 |
| Number of frames/image | 30 |
| Number of movies | 6288 |
| Camera | Gatan K2 |
| Pixel size (Å) | 1.059 |
| Symmetry imposed | C1 |
| Initial particle images (no.) | 1,031,302 |
| Final particle images (no.) | 492,931 |
| Map resolution (Å) | 3.3 |
| FSC threshold | 0.143 |
| *Refinement* | |
| Map sharpening B factor (Å) | −62.3 |
| Model composition | |
| Chain | 6 |
| Non-hydrogen atoms | 15,790 |
| Protein residues | 2020 |
| B factors (Å²) | |
| Protein | 13.61/113.06/48.51 |
| R.m.s. deviations | |
| Bond length (Å) | 0.011 |
| Bond angles (º) | 1.425 |
| Validation | |
| MolProbity score | 1.37 |
| Clashscore | 3.74 |
| Poor rotamers (%) | 0.91 |
| Ramachandran plot | |
| Favored (%) | 96.74 |
| Allowed (%) | 2.91 |
| Disallowed (%) | 0.35 |

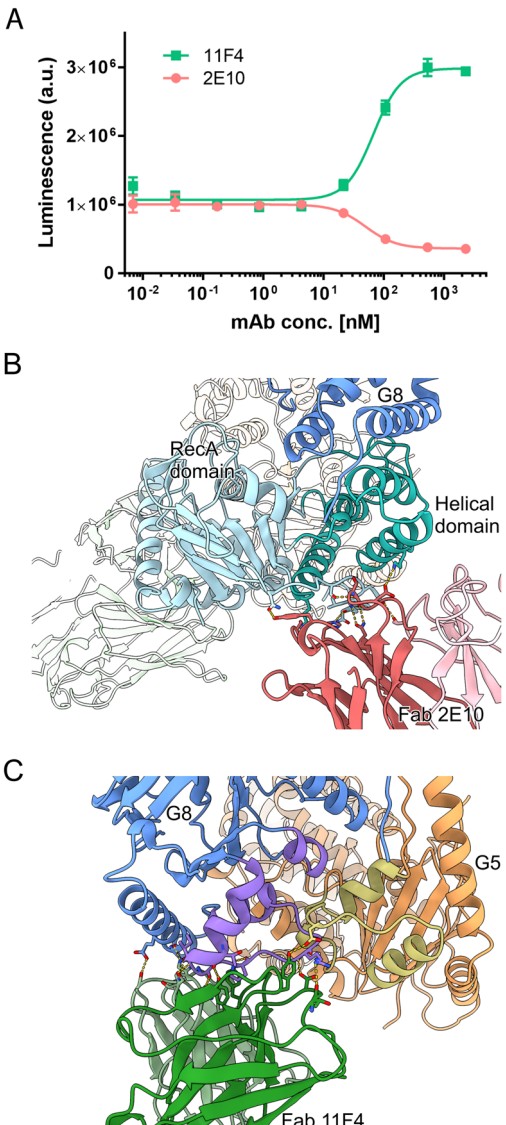

**Fig. 3 Binding of antibodies to the NBD modulates the ATPase activity of ABCG5/G8. A** ATPase activity of ABCG5/G8 in the presence of mAb 2E10 or 11F4. mAb 2E10 decreases the ATPase activity of ABCG5/G8 with an IC₅₀ of 49.4 nM. mAb 11F4 increases the ATPase activity of ABCG5/G8 with an EC₅₀ of 67.2 nM. **B** The interaction interface between Fab 2E10 and ABCG8 NBD. The helical and RecA domain of ABCG8 NBD are colored in teal and cyan, respectively. **C** The interaction interface between Fab 11F4 and ABCG5/G8. The three-helix bundle with $_{296}$NPFDFYMD$_{303}$ motif in ABCG5 and $_{316}$NPADFYVD$_{323}$ motif in ABCG8 are colored in yellow and purple, respectively.

dimer interfaces (Fig. 3C). The Fab 11F4 interface is dominated by a set of hydrogen bond interactions, where Asp32, Thr30, Tyr51, and Tyr104 from the heavy chain and Ser27, His92, His93, and Ser95 from the light chain make hydrogen bonds with residues in the C-terminus region of the NBD from ABCG8 (Supplementary Fig. 7C, D). Besides, side chain of Tyr54 from the heavy chain make a hydrogen bond with Glu293 in the C-terminus region of the NBD from ABCG5 (Supplementary Fig. 7C). Remarkably, Fab 11F4 epitopes are adjacent to the salt bridge pairs formed between the NPXDFXXD motifs of ABCG5 and ABCG8. The binding of Fab 11F4 likely rigidifies the secondary structures across the NPXDFXXD motifs, resulting in a stable closed NBD dimer interface between ABCG5 and ABCG8. Conceivably, a stable closed NBD dimer reduces the energy barrier to form a productive NBS, thus increasing the ATP hydrolysis efficiency. It is worth noting that mutations in NPXDFXXD regions in ABCG1 attenuated cholesterol efflux[24], suggesting that destabilizing the NBD dimer interface negatively affects the ATPase and transporter activity.

## Discussion
Cholesterol is a major structural component of plasma membrane and a key precursor of steroid hormone, vitamin D, and bile acid. The ABCG5/G8 heterodimer mediates the excretion of sterols in the liver and intestines, thus playing an important role in the whole-body homeostasis of cholesterol. To gain insight into the structural basis of the sterol/cholesterol transport cycle by ABCG5/G8, we developed antibodies against ABCG5/G8 to facilitate structure determination by cryo-EM. Unexpectedly, these two antibodies showed opposite effects on ABCG5/G8 ATPase activity and provide valuable tools to understand the structural mechanism of the transport cycle. While the cryo-EM structure is generally similar to the crystal structure that was solved at a lower resolution, the cryo-EM structure provides more details overall and demonstrates much better density for the NBDs. Analysis of the ABCG5/G8 cryo-EM structure and the interactions between the Fab fragments with the NBDs provide new insights into functional role of individual structure motifs.

The NBD from opposing ABCG5 and ABCG8 of the cryo-EM structure stay connected in the nucleotide-free state. Similar contact was also observed in the previously reported crystal structure of ABCG5/G8[11] and the cryo-EM structures of substrate-bound inward-facing, nucleotide-bound outward-facing, and inhibitor-bound ABCG2[12–15]. While the significance of a NPXFDXXD motif in cellular cholesterol transport assay was first described in ABCG1[24], our structure illustrates the molecular details of the interactions within this motif. Two pairs of highly ordered salt bridge interactions between the opposing NPXFDXXD motifs are arranged in a zipper-like fashion (Fig. 2F). These salt bridges help to maintain the two NBDs in close contact during the transport cycle and reduce the entropy penalty for the NBD to cycle between open and close conformation. We speculate that, by stabilizing the NBD contact, the overall efficiency for both ATPase activity and transporter cycle is increased. It is worth noting that similarly connected NBD is found in bacterial ABC exporters such as MalFGK2[25–28]. In our structure, Fab 11F4 simultaneously engages both NBDs from ABCG8 and ABCG5 and increases the ATPase activity. Interestingly, the cryo-EM structure of ABCG5/G8 reveals that the N-terminus of ABCG8 crosses over to the NBD of ABCG5 and is also part of the dimer interface (Fig. 2C). Genetic studies identified that a point mutation Asp19His in ABCG8 increases risk of gallstones, which is formed by supersaturated cholesterol in bile[10,29,30]. The first residue in ABCG8 observed in the cryo-EM structure is Phe28. While the exact location of Asp19 is not resolved in the structure, it is interesting to note that several acidic patches in ABCG5 are in proximity to Phe28. It is thus likely that Asp19His mutation might increase the ATPase and transporter efficiency by stabilizing the dimer interface.

Both the crystal and cryo-EM structures of ABCG5/G8 represent a nucleotide-free, inward-facing conformation for sterol transport. In the crystal structure of ABCG5/G8[11], electron density features representing a possible cholesterol moiety were identified in a "vestibule" formed by transmembrane helices (TMH) 1–2 of one TMD and TMH 4–6 of the opposing TMD[11]. We searched these TMH regions in the cryo-EM map but did not find noticeable features. Recently, the cryo-EM structure of ABCG2 solved in the presence of its substrate E1S (PDB: 6HCO) showed that the substrate-binding cavity is defined by hydrophobic residues from the TM2 and TM5 of opposing

protomers[12]. Surprisingly, the TMD in the cryo-EM structure of ABCG5/G8 superposes very well with the TMD of substrate-bound ABCG2, with RMSD of 2.1 Å over 440 residues. Because of the evolution conservation between ABCG2 and ABCG5/G8 and similar hydrophobic nature of their substrate, we speculate that the same pocket defined in ABCG2 might be used by ABCG5/G8 for sterol binding (Fig. 4A). While most of the pocket-lining residues are similarly hydrophobic amino acids (Fig. 4B), there are a few key differences between ABCG2 and ABCG5/G8. As shown in the sequence alignment, in contrast to the branched side chain residues in ABCG2, several key positions in ABCG5 and ABCG8 are replaced with either phenylalanine or tyrosine (Fig. 4C). Both phenylalanine and tyrosine have large aromatic side chains that shapes the binding site to accommodate flat molecules, such as cholesterol. While ABCG2 is a poly-specificity transporter that can bind structurally diverse substrate, ABCG5/G8 is highly specific for sterols, particularly plant and shellfish sterols[5,8,31]. These key residue differences in the substrate pocket are probably the basis for construction of a more selective pocket that prefers the flat four-ring steroid core structure.

The transport cycle of ABCG2 was elucidated by multiple high-resolution cryo-EM structures, which provides a foundation to understand the ATP-driven substrate transport by ABCG family. Given the overall similar molecular architecture and sequence similarity (48 and 44% sequence similarities between ABCG2 and ABCG5 and ABCG8, respectively)[32], we believe that ABCG5/G8 adopts a similar mechanism for sterol transport as ABCG2. Here, we illustrate how the two antibodies affect the ATPase activity by engaging specific conformations of ABCG5/G8 in a schematic drawing (Fig. 5). In the resting state (state I), the TMDs are facing inward, exposing a cavity for substrate binding. Substrate binding induces conformational changes to form the closed NBD dimer (state II). When ATP binds to the closed NBD dimer, RecA domains rotate around the pivot point formed by the salt bridges, and the helical domains rotate to the neighboring RecA domain to sandwich the ATP molecular by the Walker A and signature motifs. These rotational movements are essential to deliver a "power stroke" to the TMDs to extrude the substrate to either the extracellular space or the outer leaflet of the membrane (state III). Then, ATP hydrolysis breaks the connection between RecA and helical domains from the opposing half transporters. Finally, the NBDs open by rotating back around the pivot point formed by

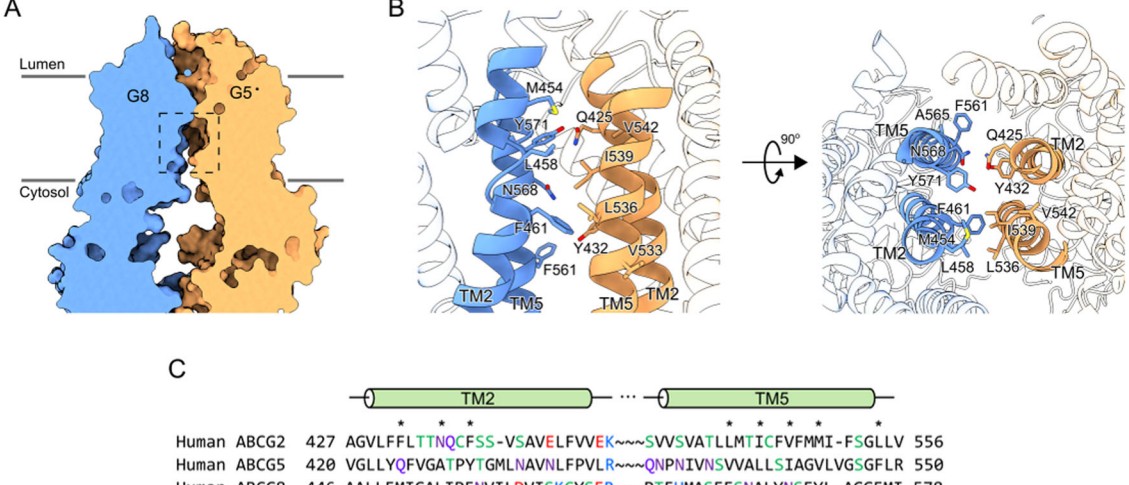

**Fig. 4 Putative cholesterol-binding pockets in ABCG5/G8. A** Cross-section of the transmembrane region of ABCG5/G8 shows a putative cholesterol-binding pocket (rectangle in dashed line). **B** Zoom-in view of the putative cholesterol-binding site shows that it is constructed by TM2 and TM5 of ABCG5 and ABCG8. Amino acid residues likely important for cholesterol binding are shown as sticks. **C** Sequence alignments of TM2 and TM5 between ABCG2, ABCG5, and ABCG8. Amino acid residues likely important for cholesterol binding are marked by an asterisk.

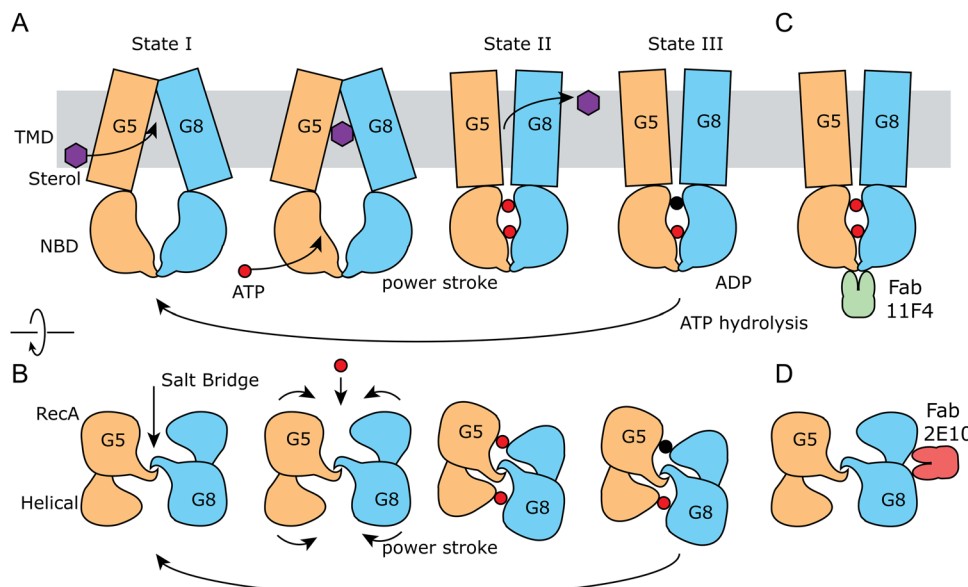

**Fig. 5 Schematic model of sterol transport by ABCG5/G8 and Fab binding. A** The sterol transport cycle of ABCG5/G8 viewed from the side of the membrane. **B** Conformational changes of NBDs in the transport cycle of ABCG5/G8 viewed from the side of cytosol. Detailed descriptions of the transport cycle of ABCG5/G8 can be found in the "Discussion" section. **C** Fab 11F4 (green) interacts with the dimer interface of the NBDs, which stabilizes the dimerization of NBDs favoring ATP hydrolysis. **D** Fab 2E10 (red) interacts with both the RecA and helical domains of ABCG8, which restricts the relative movement between the RecA and helical domains and decreases ATP hydrolysis.

the salt bridges and reset the overall conformation to state I. In this schematic drawing, the movement between RecA and helical domain is critical. Fab 2E10 restricts such movement and thus inhibits ABCG5/G8 ATPase activity. On the other hand, Fab 11F4 activates ABCG5/G8 by stabilizing the NBD dimer interactions.

An extensive list of disease-causing alleles has been identified in ABCG5/G8 (Supplementary Fig. 8) but pharmaceutic interventions of ABCG5/G8 have been very challenging due to poor understanding of the transport cycle. Compounds that potentiate the transport activity of ABCG5/G8 and accelerate sterol excretion may have profound therapeutic significance. Although the antibodies described here cannot be used as therapeutic agents due to the poor penetration of antibodies across plasma membrane to reach the intracellular epitope, these findings uncover allosteric-binding sites for novel membrane-penetrating modalities such as small molecules to modulate ABCG5/G8 activity for therapeutic development.

## Methods

**Protein expression and purification.** Human ABCG5 with a TEV protease cleavage site and 12 His tag on the C-terminal and ABC G8 was cloned into pD902 vectors and integrated into ATUM-9016 *Pichia pastoris* strain. The cells were grown in Buffered Minimal Glycerol (BMG) medium (100 mM potassium phosphate pH 6.0, 1.34% w/v yeast nitrogen base without amino acid and with ammonium sulfate, and 4 e−5% w/v biotin) and induced the expression in Buffered Minimal Methanol(BMM) medium (200 mM potassium phosphate pH 60, 1.34% (w/v) yeast nitrogen base, 4 e−5% (w/v) biotin, 0.7% (v/v) methanol) at 30 °C. The cell was harvested by centrifugation at $4000 \times g$ for 10 min and lysed using microfluidizer. Membranes were isolated by centrifugation at $200,000 \times g$ for 1 h, then resuspended in half lysis volume of the membrane prep buffer (20 mM Tris pH 8, 100 mM NaCl, 2 mM dithiothreitol (DTT), 10% v/v glycerol). In all, 1.6 M NaCl was added to the membrane resuspension. The membranes were centrifuged again at $200,000 \times g$ for 40 min. The pellet was resuspended in the membrane prep buffer in half lysis volume again. Then, membranes were extracted by adding 2% w/v DDM and stirring for 30 min at 4 °C. The extract was further centrifuged at $200,000 \times g$ for 40 min. The supernatant was isolated and 40 mM imidazole was added. The protein was purified using a HisTrap 5 mL column with 40 mM imidazole binding, 50 mM imidazole wash (60 mL), and 400 mM imidazole elution (10 mL). The purified protein was further polished on Superdex 200 increase 10/300 column in a buffer containing 20 mM Tris, pH 7.9, 150 mM NaCl, 0.02% DDM, and 1 mM DTT. The purified protein was aliquoted and stored at

−80 °C before use. Antibodies against ABCG5/G8 were produced at the antibody core facility at Oregon Health Science University. The Fab fragments were produced by papain cleavage[33] and cleaved Fabs were further purified by gel filtration chromatography (GE Healthcare). ABCG5/G8 was incubated with Fab 2E10 in molar ratio of 1:1.25 for 30 min on ice and the complex of ABCG5/G8 with Fab 2E10 was purified on Superdex 200 increase 10/300 column. The purified complex was incubated with Fab 11F4 in molar ratio of 1:1.25 for 30 min on ice and the ternary complex of ABC G5/G8 with Fab 2E10 was purified on Superdex 200 increase 10/300 column freshly before reconstitution into saposin A-based nanodiscs. Saposin A protein was expressed and purified as described[34]. The saposin A-based nanodiscs were prepared using a modified protocol as described by Frauenfeld et al.[34].

**Surface plasmon resonance.** SPR-binding studies were performed using a Biacore T200 system (GE Healthcare). A Biacore CM5 chip was first functionalized with goat-anti mouse IgG, Fc fragment-specific antibody (Jackson ImmunoResearch, PA) for capturing mouse anti-ABCG5/G8 antibodies for binding characterization. The capturing level of anti-ABCG5/G8 antibodies were ~100 Ru. ABCG5/G8 in DDM detergent micelle was tested in multicycle kinetics with concentration ranging from 6 pM to 100 nM. The SPR assay was performed at 50 μL/min flow rate with 180 s of association and 1800 s of dissociation.

**ATPase assay.** ATPase assay was performed using the ADP-Glo Kinase Assay Kit (Promega) with previously established protocol. Basically, 260 nM ABCG5/G8 and Fab in different concentrations were added to 10 μL reaction mixtures containing 50 mM Tris pH 7.4, 0.02% DDM, 1 mM DTT, 25 mM NaCl, 4 mM MgCl₂, 3 mM sodium cholate, 0.5 mM 1,2-dioleoyl-sn-glycero-3-phospho-L-serine, and 0.06 mM cholesterol. The reaction was started by adding 25 μM ATP and incubated at 37 °C for 2 h. To detect the ADP reaction product, the reaction mixture was cooled down at room temperature for 30 min. Subsequently, 5 μL ADP-Glo™ reagent was added to the mixture and incubated at room temperature for 1 h. In all, 10 μL kinase detection reagent was added and incubated at room temperature for 1 h. The luminescence from the reaction was detected using EnVision Multimode Plate Reader in 0.25 s/int with a US-Lumi filter.

**Cryo-EM sample preparation and data collection.** For grid preparation, 2.5 μL of ABCG5/ABCG8 complex in saposin A nanodiscs (~2 mg/mL, supplemented with 0.5 mM fluorinated Fos-choline-8) was applied to glow-discharged Quantifoil R1.2/1.3 300 mesh grids. The grids were blotted for 3 s at 20 °C/100% relative humidity and then plunge frozen with liquid ethane using a FEI Vitrobot Mark IV. For data collection, the grids were transferred to a Titan Krios microscope operated at 300 kV and imaged with a Gatan K2 camera. Dose fractionated movies were recorded at a nominal magnification of ×130,000 (corresponding to a physical pixel size of 1.059 Å). Six-second exposures were divided into 0.2-s movie frames and recorded in super-resolution mode, with a total dose of 48 electrons/Å². MotionCor2 was

used to align the movie frames to generate dose-weighted, 2× binned images[35], and CTF estimation was carried out using Gctf[36].

**Data processing**. The aligned images were visually inspected to remove bad images with contamination or with obvious drift, resulting in a final set of 6288 images. About 1000 particles were manually picked in Relion-3.0 to generate reference-free two-dimensional (2D) class averages and the best class averages were used as templates for autopicking in Relion-3.0[37]. A total of 1.03 million particles were picked and downsized to 2.118 Å/pixel and then subjected to 2 rounds of reference-free 2D classification to remove bad particles and contaminants. After 2D classification, an initial 3D model was automatically generated in Relion-3.0 and used for 3D classification. The 3D class with high-resolution features was selected and the particles belonging to this class were re-extracted at 1.059 Å/pixel. An additional 3D classification step resulted in a final class containing 492,931 particles, which were used for 3D auto-refinement, CTF refinement, and Bayesian polishing in Relion-3.1. The final resolution of the 3D map was estimated to be 3.3 Å based on the FSC 0.143 criterion. Real-space refinement is carried out in Phenix[38] and the structure modeling is performed in COOT[39]. All structural figures are prepared with Pymol (Schrödinger, CA) and UCSF ChimeraX[40].

**Statistics and reproducibility**. No statistical method was used to determine the sample size, and the experiments were not randomized, because only biochemical experiments are involved in this paper. The summary of the cryo-EM data collection, refinement, and validation statistics are shown in Table 1.

**Reporting summary**. Further information on research design is available in the Nature Research Reporting Summary linked to this article.

## Data availability

The density map and model coordinate were submitted in PDB and EMDB with IDs: PDB ID: 7JR7; EMDB ID: EMD-22443. The raw data underlying Fig. 1a–d and 3a can be found in Supplementary Data 1. Other data are available by reasonable request.

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

## Acknowledgements

We thank Chen Xu and Kangkang Song at the UMass cryo-EM facility for help with electron microscopic data collection, Daniel Cawley at Oregon Health Science University for assistance on antibody generation, and Athena Sudom for critical discussions and helpful comments.

## Author contributions

M.Z., X.M., and Z.W. conceived the project. J.L. and H.Z. generated protein samples, X.M. prepared materials for cryo-EM imaging. X.Y. prepared cryo-EM grids and collected and analyzed cryo-EM data. C.-S.H. and X.M. analyzed cryo-EM data and built the models. A.V. and Q.C. performed SPR experiments. H.Z. conducted ATPase assays. C.-S.H., H.Z., and X.M. wrote the manuscript with input from all authors.

## Competing interests

The authors declare no competing interests.
