## [Peer Review File · Communications Biology]

Reviewers' comments:

Reviewer #1 (Remarks to the Author):

Huang et al present the structure of the sterol transporter ABCG5/8 in complex with two monoclonal antibody fragments which have specific and different effects on G5/8 function.

Overall, the study provides new insight into G5/8 structure, possibly allowing the correction of registry errors in the NBDs which gives better understanding of structure-function relationships in this transporter. The affinity of the antibodies and their effect on ATPase activity are well studied. The manuscript is clearly written.

Despite these positive points, there are some issues that could be resolved or explained.

- 1) The authors show the structure of G5/8 with 2 different antibody fragments. However, for G2 e.g., it was eventually possible to get $<4 \text{ \AA}$ cryo-EM structures without antibody fragments. Did the authors attempt structure determination without these fragments?
- 2) Similarly, did they attempt structure solution of the transporter with either one (rather than both) mAB fragments?
- 3) The authors claim they resolve registry errors in the G5/8 crystal structure. Did they reevaluate the G5/8 crystal structure electron density? The authors should provide supplementary figures to strengthen their claim.
- 4) The biochemical studies are rather limited (only ATPase activity is studied, not transport). Study of transport could give additional insight into the mechanism of the antibodies.
- 5) The authors should consider making a supplementary figure with the known mutations in G5/8 and plotting them on the structure.
- 6) The authors claim e.g. in the abstract that mAB 11F4 stabilizes NBD dimer formation and mAB 2E10 restricts the relative movement between the RecA and helical domain. Without structural or functional data, these claims should be qualified with a word such as "likely", "probably" or "possibly".
- 7) The authors should detail the protein complex sample preparation as used for cryo-EM (how was the G5/8 in complex with two mAB fragments prepared), in the methods (as well).
- 8) Supplementary figure 1: there should be a scale bar in a. The authors should mention the size of the boxes (in \AA) in b
- 9) As I understand it, the authors purified the protein from *Pichia pastoris* (this can be briefly mentioned in the methods even if referring to the original reference). Does *P. pastoris* produce cholesterol in significant amounts and if not could the authors speculate as to what other sterol molecule might be in their structure?
- 10) Did the authors use particle polishing and local CTF correction to improve their maps?

Minor points

Line 73: Signal-noise-ratio  signal-to-noise-ratio

Line 75: protein  proteins

Line 238: out of cell  out of the cell

Line 349: Relion3.0  change and be consistent

Line 359: withPymol(Schrodinger, CA)  with Pymol (Schrödinger, CA)

Reviewer #2 (Remarks to the Author):

Huang et al., report the cryo-EM structures of the human ABC transporter ABCG5/G8 (G5/G8) in FABs bound inward-facing state. While the crystal structure of inward-facing G5/G8 has been described previously (Lee et. al 2016), the new cryo-EM structure of inward-facing G5/G8 described here was determined in a significantly higher quality. With this new cryo-EM map, the authors were able to fix a few modelling errors in the previous crystal structure, and revealed several novel structural features of inward-facing G5/G8. Particularly, the authors found out that the NPXFDXXD motifs from G5 and G8 interact with each other, thereby contributing to the dimerization of NBDs. In addition, the authors performed structural and functional analysis of the

FAB bindings, and provided reasonable structural explanations as to how these two FABs have opposite effects on the ATP activity of G5/G8.

Overall, the cryo-EM and biochemical works looks solid. By elucidating a more precise structure of G5/G8 in an inward-facing conformation, this manuscript advances our knowledge and understanding of the transporting mechanism of G5/G8 to some extent. This FAB bound structure will also facilitate the design of better FABs for modulating the activity of G5/G8. Considering these, this work can potentially make certain impact on the field of ABC transporter, but I think the present manuscript would be strengthened by addressing the following concerns.

- (1) The two FABs (2E10 and 11F4) have completely opposite effects on the ATP activity of G5/G8, suggesting that different types of FAB binding may trap G5/G8 in distinct conformations. I don't think it is necessary to collect new cryo-EM data, but I wonder if authors have obtained some structural information for the structure of G5/G8 with only 2E10 or 11F4 bound. Even those structures were resolved at lower resolution, some comparisons could be conducted to understand the protein dynamics.
- (2) The authors made a strong claim that the modeling of G8 loop 28-56 and G8 loop 306-355 is not correct in the previous crystal structure of G5/G8. To convince the readers this is the case, the author should show the fitting of these two loops into cryo-EM density more clearly (i.e., show them in a larger figure panel, in different orientation, and with better resolution.)
- (3) The discussion of the putative sterol binding site between TM2 and TM5 is very speculative, as this is mainly based on the structural comparison between G2 and G5/G8. No cryo-EM density of sterol could be observed at this site, and there is no any biochemical results indicating that this site of G5/G8 can bind sterol. The authors should consider altering the writing of this part and move it to discussion.
- (4) "During the model building process, we identified a density feature in a surface pocket on ABCG5 that is reminiscent of the shape of a cholesterol molecule". While the authors provided a figure showing this lipid density, it remains unclear to me why this density was ascribed to cholesterol. Judging from Figure 4D, this density appears to be thin and elongated, which is quite different to the typical shape of a cholesterol. Do the authors have other evidences to support such lipid assignment? If the authors can't provide other evidences, it would be necessary to remove this supposition from the manuscript to avoid any misleading.
- (5) The scale bar is missing in Supplementary figure 1A.
- (6) It would be clearer if the authors could indicate the locations of the models shown in Figure 1C-F in the overall structure of G5/G8.

Rebuttal letter

Here are the point by point response to the reviewers' comments. Our response is in blue.

Reviewer#1:

1) *The authors show the structure of G5/8 with 2 different antibody fragments. However, for G2 e.g., it was eventually possible to get <4 Å cryo-EM structures without antibody fragments. Did the authors attempt structure determination without these fragments?*

We appreciate the interests from both reviewers on the cryo-EM structure of ABCG5/G8 alone. Indeed, we worked hard to try to solve the structure without Fab in the beginning. Unfortunately, the transporter showed strong preferred orientation in EM-grid and we were not able to collect good-quality data on these grids. While we were working on the structure with no Fab, we also initiated the antibody screening campaign since we expect that the relatively small molecular weight of the ABCG5/G8 dimer and the pseudo-symmetry between ABCG5 and ABCG8 half transporter may pose a challenge for high resolution cryo-EM structure. The sample with two antibody fragments significantly improved the preferred-orientation problem and allowed us to solve the structure at high resolution.

2) *Similarly, did they attempt structure solution of the transporter with either one (rather than both) mAB fragments?*

We thank the reviewers' suggestion. Unfortunately, due to the limited access to the cryo-EM facility, we were not able to perform structural studies of ABCG5/G8 with only one Fab. We agree with the reviewer that it would be interesting to understand the effect of individual Fabs on the conformation of the transporter, given the distinctive effects of the two antibodies on ATPase activity. It is possible that mAb 2E10 clamped the helical and RecA domain and prevented conformational changes in the transport cycle. Thus mAb 2E10-only binding structure may resemble our current cryo-EM structure. On the other hand, mAb 11F4 may induce a conformational change in the NBD that favors ATPase activity, or the binding of mAb 11F4 may stabilize an intermediate conformation during the transport cycle. To capture such conformation, we plan to test structures of ABCG5/G8 bound with Fab 11F4 with either non-hydrolyzable ATP or with a ABCG5 E218→Q mutant to slow down the ATPase cycle. We hope to spend more resources to study this topic and report the findings in a future publication.

3) *The authors claim they resolve registry errors in the G5/8 crystal structure. Did they reevaluate the G5/8 crystal structure electron density? The authors should provide supplementary figures to strengthen their claim.*

We thank the reviewer for the excellent suggestion. We have re-evaluated the ABCG5/G8 crystal structure electron density. Guided by the cryo-EM structure, we rebuilt the model for the G5/8 crystal structure density and performed refinement in Phenix. The rebuilt model match reasonably better with the electron density and we have added a supplementary figure (Fig S3). In addition, there is a slight decrease in both R-free and the gap between R-free and R-work.

4) *The biochemical studies are rather limited (only ATPase activity is studied, not*

transport). Study of transport could give additional insight into the mechanism of the antibodies.

We agree with the reviewer that additional biochemical studies such as a transport assay may provide more insight into the mechanism of action of the antibodies. However, since the substrate of G5G8 is sterol, which is retained in the lipid membrane before and after transport, it is challenging to probe the distribution of sterol in the inner and outer leaflet on the membrane. With our current equipment set up, we couldn't develop a transport assay for G5G8.

5) The authors should consider making a supplementary figure with the known mutations in G5/8 and plotting them on the structure.

We thank the reviewer for the suggestion. We have added a supplementary figure (Fig S7) with the known mutations.

6) The authors claim e.g. in the abstract that mAB 11F4 stabilizes NBD dimer formation and mAB 2E10 restricts the relative movement between the RecA and helical domain. Without structural or functional data, these claims should be qualified with a word such as "likely", "probably" or "possibly".

We agree with the reviewer on the lack of functional data on mAb 11F4 and 2E10. We have revised our manuscript based on your suggestion.

7) The authors should detail the protein complex sample preparation as used for cryo-EM (how was the G5/8 in complex with two mAB fragments prepared), in the methods (as well).

Thank you for your suggestions. We have revised our manuscript based on your suggestion.

8) Supplementary figure 1: there should be a scale bar in a. The authors should mention the size of the boxes (in Å) in b

We have revised the figure based on your suggestion.

9) As I understand it, the authors purified the protein from Pichia pastoris (this can be briefly mentioned in the methods even if referring to the original reference). Does P. pastoris produce cholesterol in significant amounts and if not could the authors speculate as to what other sterol molecule might be in their structure?

The main sterol produced by yeast is ergosterol (<https://doi.org/10.1016/j.bbamem.2014.03.012>). The cryo-EM structure represents a nucleotide-free, inward-facing conformation and we do not expect to find substrate (preferably plant sterol) in the active site. We plan to make ABCG5 E218→Q mutation to trap the substrate in follow-up studies.

10) Did the authors use particle polishing and local CTF correction to improve their maps?

We used particle polishing and local CTF correction at the final stage using Relion 3.1 to achieve high resolution.

Minor points

Line 73: Signal-noise-ratio  signal-to-noise-ratio

Line 75: protein  proteins

Line 238: out of cell  out of the cell

Line 349: Relion3.0  change and be consistent

Line 359: withPymol(Schrodinger, CA)  with Pymol (Schrödinger, CA)

Thank you for your comments. We have revised the manuscript accordingly.

Reviewer #2:

(1) The two FABs (2E10 and 11F4) have completely opposite effects on the ATP activity of G5/G8, suggesting that different types of FAB binding may trap G5/G8 in distinct conformations. I don't think it is necessary to collect new cryo-EM data, but I wonder if authors have obtained some structural information for the structure of G5/G8 with only 2E10 or 11F4 bound. Even those structures were resolved at lower resolution, some comparisons could be conducted to understand the protein dynamics.

Thank you to both reviewers for raising the question on the structure of ABCG5/G8 with one Fab. We are indeed very interested to perform further experiment using one Fab fragment in combination with active site mutants. However, due to the current pandemic situation and our extremely limited access time to EM, we were not able to pursue ABCG5/G8 complex structure with one Fab.

(2) The authors made a strong claim that the modeling of G8 loop 28-56 and G8 loop 306-355 is not correct in the previous crystal structure of G5/G8. To convince the readers this is the case, the author should show the fitting of these two loops into cryo-EM density more clearly (i.e., show them in a larger figure panel, in different orientation, and with better resolution.)

Thank you for the excellent question. As further proof of our argument, we have re-evaluated the ABCG5/G8 crystal structure electron density. Guided by the cryo-EM structure, we rebuilt the model for the G5/8 crystal structure density and performed refinement in Phenix. The rebuilt model match reasonably better with the electron density and we have added a supplementary figure (Fig S3). In the figure, we compare side by side the density and model of selected regions of loop 28-56 and 306-355.

(3) The discussion of the putative sterol binding site between TM2 and TM5 is very speculative, as this is mainly based on the structural comparison between G2 and G5/G8. No cryo-EM density of sterol could be observed at this site, and there is no any biochemical results indicating that this site of G5/G8 can bind sterol. The authors should consider altering the writing of this part and move it to discussion.

Thank you for the suggestion. We revised our manuscript based on your suggestion.

(4) "During the model building process, we identified a density feature in a surface pocket on ABCG5 that is reminiscent of the shape of a cholesterol molecule". While the authors provided a figure showing this lipid density, it remains unclear to me why this density was ascribed to cholesterol. Judging from Figure 4D, this density appears to be thin and elongated, which is quite different to the typical shape of a cholesterol. Do the authors have other evidences to support such lipid assignment? If the authors can't

provide other evidences, it would be necessary to remove this supposition from the manuscript to avoid any misleading.

Thank you for the suggestion. We removed this section from our manuscript based on your suggestion.

(5) The scale bar is missing in Supplementary figure 1A.

We revised the figure S1A based on your suggestion.

(6) It would be clearer if the authors could indicate the locations of the models shown in Figure 1C-F in the overall structure of G5/G8.

Thank you for the excellent suggestions. We revised Figure 2 in our manuscript based on your suggestion.

REVIEWERS' COMMENTS:

Reviewer #1 (Remarks to the Author):

The authors convincingly answered all questions.

I just have two more remarks:

1. Do they mean Relion 3.0 or 3.1 (and for which parts of the processing) – it is not clear as it differs between rebuttal letter and paper? They should write this correctly where applicable.
2. The authors should deposit the revised X-ray coordinates as well (as a suggestion, coordinating with the researchers who originally collected the X-ray data).

Reviewer #3 (Remarks to the Author):

The authors have address most of my points raised in the initial review by improving the presentations. The structural work present here is of higher quality than previous crystal structure, so it indeed provides more accurate structural information for ABCG5/G8, which advances this field. I therefore support its publication at Communication Biology. It is still worth the effort to further investigate the structure of G5/G8 with single FAB bound in the future.